# Transferrin Saturation/Hepcidin Ratio Discriminates *TMPRSS6*-Related Iron Refractory Iron Deficiency Anemia from Patients with Multi-Causal Iron Deficiency Anemia

**DOI:** 10.3390/ijms23031917

**Published:** 2022-02-08

**Authors:** Hilde van der Staaij, Albertine E. Donker, Dirk L. Bakkeren, Jan M. J. I. Salemans, Lisette A. A. Mignot-Evers, Marlies Y. Bongers, Jeanne P. Dieleman, Tessel E. Galesloot, Coby M. Laarakkers, Siem M. Klaver, Dorine W. Swinkels

**Affiliations:** 1Translational Metabolic Laboratory, Department of Laboratory Medicine, Radboud University Medical Center (Radboudumc), 6525 GA Nijmegen, The Netherlands; h.vanderstaaij@sanquin.nl (H.v.d.S.); Albertine.Donker@radboudumc.nl (A.E.D.); coby.laarakkers@radboudumc.nl (C.M.L.); siem.klaver@radboudumc.nl (S.M.K.); 2Máxima Medical Center (MMC), Department of Pediatrics, 5504 DB Veldhoven, The Netherlands; 3 Máxima Medical Center (MMC), Department of Clinical Chemistry, 5504 DB Veldhoven, The Netherlands; d.bakkeren@mmc.nl; 4Máxima Medical Center (MMC), Department of Gastroenterology, 5504 DB Veldhoven, The Netherlands; j.salemans@mmc.nl; 5Máxima Medical Center (MMC), Emergency Department, 5504 DB Veldhoven, The Netherlands; l.evers@mmc.nl; 6Máxima Medical Center (MMC), Department of Gynecology, 5504 DB Veldhoven, The Netherlands; m.bongers@mmc.nl; 7Maastricht University Medical Center, Department of Gynecology, 6229 HX Maastricht, The Netherlands; 8Máxima Medical Center Academy, Máxima Medical Center (MMC), 5504 DB Veldhoven, The Netherlands; j.dieleman@mmc.nl; 9Department for Health Evidence, Radboud Institute for Health Sciences, 6500 HB Nijmegen, The Netherlands; Tessel.Galesloot@radboudumc.nl; 10Hepcidinanalysis, Translational Metabolic Laboratory, Geert Grooteplein 10, 6525 GA Nijmegen, The Netherlands

**Keywords:** IRIDA, iron deficiency anemia, *TMPRSS6*, hepcidin, transferrin, transferrin saturation/hepcidin ratio

## Abstract

Pathogenic *TMPRSS6* variants impairing matriptase-2 function result in inappropriately high hepcidin levels relative to body iron status, leading to iron refractory iron deficiency anemia (IRIDA). As diagnosing IRIDA can be challenging due to its genotypical and phenotypical heterogeneity, we assessed the transferrin saturation (TSAT)/hepcidin ratio to distinguish IRIDA from multi-causal iron deficiency anemia (IDA). We included 20 IRIDA patients from a registry for rare inherited iron disorders and then enrolled 39 controls with IDA due to other causes. Plasma hepcidin-25 levels were measured by standardized isotope dilution mass spectrometry. IDA controls had not received iron therapy in the last 3 months and C-reactive protein levels were <10.0 mg/L. IRIDA patients had significantly lower TSAT/hepcidin ratios compared to IDA controls, median 0.6%/nM (interquartile range, IQR, 0.4–1.1%/nM) and 16.7%/nM (IQR, 12.0–24.0%/nM), respectively. The area under the curve for the TSAT/hepcidin ratio was 1.000 with 100% sensitivity and specificity (95% confidence intervals 84–100% and 91–100%, respectively) at an optimal cut-off point of 5.6%/nM. The TSAT/hepcidin ratio shows excellent performance in discriminating IRIDA from *TMPRSS6*-unrelated IDA early in the diagnostic work-up of IDA provided that recent iron therapy and moderate-to-severe inflammation are absent. These observations warrant further exploration in a broader IDA population.

## 1. Introduction

Matriptase-2, a transmembrane serine protease encoded by *TMPRSS6,* plays a key role in downregulating hepcidin expression through modulation of the BMP-SMAD pathway when iron stores are low [1,2,3,4]. Pathogenic *TMPRSS6* variants result in impaired matriptase-2 function, leading to inappropriately high plasma hepcidin levels in relation to body iron status [5,6]. Since hepcidin impairs intestinal iron absorption and recycling by inhibiting ferroportin-mediated iron export from enterocytes and macrophages, patients with *TMPRSS6*-related iron refractory iron deficiency anemia (IRIDA) develop microcytic anemia with remarkably low transferrin saturation (TSAT), low–normal ferritin levels, and a poor response to oral iron treatment [5,7,8,9,10,11,12].

Usually, biallelic pathogenic *TMPRSS6* variants are found in IRIDA patients and therefore the disease is considered to be inherited in an autosomal recessive way. However, anecdotal data are available of phenotypically affected IRIDA patients in whom only a heterozygous pathogenic *TMPRSS6* variant was found [5,12], corroborating our observations [11]. Biallelic affected patients typically present in childhood, while monoallelic affected patients generally present later in life with a milder phenotype regarding the severity of microcytic anemia [7,10,11].

Diagnosing *TMPRSS6*-related IRIDA can be challenging, as the disorder is phenotypically and genotypically heterogeneous. In our case series, the median age at the time of evaluation for IRIDA was 15 years (range 1–40 years) and 41 years (range 32–48 years) for biallelic and monoallelic affected patients, respectively. Most patients undergo extensive diagnostic workup, including invasive diagnostic tests (e.g., gastrointestinal endoscopy), before IRIDA is diagnosed and effective treatment (most often parenteral iron) can be initiated [11]. Therefore, we sought to develop a tool to assist clinicians in recognizing and differentiating *TMPRSS6*-related IRIDA from other common causes of iron deficiency anemia (IDA) to ensure timely diagnosis and prevent unnecessary invasive diagnostic workup. Since the cardinal feature of IRIDA is the presence of discrepantly high hepcidin levels relative to a low body iron status in general, and a low circulating iron pool in particular, it has been suggested that the ratio between TSAT and hepcidin is a promising tool in diagnosing IRIDA [12,13,14]. Moreover, our previous small study indicated that the TSAT/hepcidin ratio was able to discriminate between biallelic and monoallelic IRIDA patients and between monoallelic IRIDA patients and their phenotypically unaffected relatives with the same heterozygous *TMPRSS6* variant, even after parenteral iron therapy had been given [11].

In the present study, we assessed the ability of the TSAT/hepcidin ratio to distinguish biallelic and monoallelic affected IRIDA patients from patients with IDA due to other causes, such as hypermenorrhea, gastrointestinal hemorrhage, and malabsorption, who had no signs of moderate-to-severe inflammation and who have not received recent iron therapy to preclude significant modulation of the hepcidin regulatory pathway that could affect the ratio between TSAT and hepcidin [8,15,16,17,18,19,20]. We aimed to establish a cut-off point for the TSAT/hepcidin ratio to discriminate between both groups with high sensitivity and specificity, using hepcidin values obtained by a standardized assay that allows for the direct comparison of hepcidin concentrations and their ratios with TSAT results obtained by other validated and standardized hepcidin assays worldwide [21,22]. Our observations pave the way for diagnosing *TMPRSS6*-related IRIDA in a more timely and efficient manner in patients with unexplained IDA.

## 2. Results

### 2.1. Patient Characteristics

#### 2.1.1. IRIDA Patients

Twenty patients with *TMPRSS6*-related IRIDA (called ‘IRIDA’ hereafter) were identified in the Iron Biobank registry and fulfilled our entry criteria (Figure 1). Seventeen of the twenty patients were diagnosed between 2010 and 2015 and have been described before [11]. Three IRIDA patients were diagnosed between 2015 and 2019 and have not been described elsewhere. Clinical, biochemical, and genetic characteristics of the IRIDA group are presented in Table 1. Eleven patients, seven females and four males, carried a biallelic pathogenic *TMPRSS6* variant, while nine patients, eight females and one male, carried a monoallelic pathogenic *TMPRSS6* variant. The median age at the time of evaluation for IRIDA was 9 years (interquartile range, IQR, 6–21 years) in the biallelic group and 40 years (IQR 31–48 years) in the monoallelic group. C-reactive protein (CRP) levels at the time of TSAT/hepcidin assessment were <5.0 mg/L in eighteen (90%) IRIDA patients, and 5.4 mg/L and 8.0 mg/L in the other two IRIDA patients, respectively. All *TMPRSS6* results in the IRIDA group were classified as either class 4 (likely pathogenic) or class 5 (clearly pathogenic) variants. In the monoallelic patients, multiplex ligation-dependent probe amplification (MLPA) showed no large deletions or duplications in the second allele.

#### 2.1.2. IDA Controls

A total of 161 patients underwent diagnostic workup for IDA during the study period. Of these, 122 were excluded (Figure 1, reasons not mutually exclusive) due to recent iron therapy (<3 months), not meeting the inclusion criteria for IDA, CRP ≥ 10.0 mg/L, active inflammatory conditions, malignancy, bariatric surgery, severe kidney or liver disease, or no written informed consent. The remaining 39 subjects were included in the control group. Clinical, biochemical, and treatment characteristics of the IDA controls are presented in Table 1 and Table 2.

In the IDA group, 74% were female. The median TSAT and ferritin levels were 5.0% (IQR 3.0–7.0%) and 9.0 µg/L (IQR 6.0–14.0 µg/L), respectively. The median CRP level was 1.8 mg/L (IQR 0.7–2.5 mg/L). According to international cut-off points for body mass index (BMI) [23], 23% of the IDA controls were overweight (i.e., BMI 25.0–29.9 kg/m^2^), and 44% were obese (i.e., BMI ≥ 30.0 kg/m^2^). Comorbidity was present in 46% of the controls, with diabetes mellitus being the most prevalent condition (Table 1). Renal function was estimated in 85% of the IDA controls. Of these, 49% had an estimated Glomerular Filtration Rate (eGFR) ≥ 90 mL/min/1.73 m^2^, and 44% had a mild to moderate decrease in eGFR (i.e., >30 and <90 mL/min/1.73 m^2^).

After diagnostic workup of IDA, in 54% of the controls an underlying condition was found, of which gastrointestinal bleeding was the most prevalent cause (Table 2). However, despite extensive diagnostic evaluation (including gastrointestinal endoscopy), in 46% of the controls IDA remained unexplained. In these cases, IDA was attributed to an insufficient diet, obesity [24], or to side effects associated with drugs that increase the risk of bleeding (e.g., anticoagulants or non-steroidal anti-inflammatory drugs, NSAIDs) or decrease iron absorption (e.g., proton pump inhibitors). In subjects with unexplained IDA, the responsiveness to iron therapy was assessed (Table 2). A good response, i.e., ≥2.0 g/dL hemoglobin (Hb) increment after 3 weeks of iron therapy [25], to either oral or intravenous (IV) iron supplementation was observed in 72% of the subjects. Seventeen percent received no iron therapy because of spontaneous Hb normalization without treatment, gastrointestinal side effects after iron supplementation in the past (>3 months ago), and refusal of iron therapy. In one control the response to iron therapy was unknown.

Whole blood samples of 36 out of 39 IDA controls (92%) were available for DNA analysis of *TMPRSS6* exons. Genotyping of these samples showed one heterozygous pathogenic *TMPRSS6* variant: c.1324G>A p.Gly442Arg [5,26,27]. In functional studies, this variant showed a partial defect in hemojuvelin cleavage [6,26,28] and was therefore considered pathogenic (class 5). This subject was a 69-year-old woman who had microcytic anemia (Hb 11.8 g/dL, mean corpuscular volume, MCV, 77.0 fL) with a TSAT level of 6.0%, a hepcidin level of 0.8 nM, a ferritin level of 24.0 µg/L, and no signs of inflammation (CRP 1.6 mg/L). After gastrointestinal endoscopy, the cause of IDA remained unexplained and was attributed to the use of proton pump inhibitors and antiplatelet therapy. There was no trial of oral iron therapy, as parenteral iron treatment was directly provided by the treating physician. After IV iron treatment, a good response of Hb, MCV and TSAT was observed, which persisted at follow-up 5 months later. We considered IRIDA unlikely in this patient, as there was no recurrent anemia after 5 months without iron therapy, which does not fit the characteristics of an IRIDA phenotype. Therefore, we retained this subject in the control group for our primary analyses and we assessed in a sensitivity analysis the effect of excluding this subject on the diagnostic performance of the TSAT/hepcidin ratio.

The *TMPRSS6* sequencing results of the other IDA controls, including subjects that had not received iron therapy (*n* = 3) and ones with a poor (*n* = 1) or unknown (*n* = 1) response to iron supplementation, all showed either wildtype, class 1 (clearly not pathogenic) or class 2 (unlikely to be pathogenic) variants. In controls with missing DNA for *TMPRSS6* sequencing (*n* = 3), the cause of IDA was attributed to gastrointestinal bleeding and obesity, hypermenorrhea and obesity, and medication use (proton pump inhibitor and anticoagulant therapy, good response to oral iron supplementation), respectively.

#### 2.1.3. Comparison of Baseline Characteristics

At the time of diagnosis of IRIDA and TSAT/hepcidin assessment, biallelic and monoallelic IRIDA patients had a median age of 9 years (IQR 7–20 years) and 40 years (IQR 31–38 years), respectively. IDA controls had a median age of 62 years (IQR 52–71 years) at the time of inclusion, significantly older than the IRIDA patients (*p* < 0.001). Women were equally represented in both groups (*p* = 0.8). CRP levels at the time of TSAT/hepcidin assessment were below the upper limit of the reference range of <5.0 mg/L in 18 out of 20 IRIDA patients (90%) and in 33 out of 39 IDA controls (85%). In IRIDA patients, data on BMI levels and eGFR were not available, but we anticipated that in this group the number of persons with impaired eGFR, overweight, and obesity would probably be lower, as generally young people are more fit and athletic compared to people aged 50 years and above. In the IRIDA group, MCV levels were significantly lower and ferritin levels significantly higher in comparison with the IDA group (*p* = 0.01 and *p* < 0.001).

### 2.2. TSAT, Hepcidin Levels, and TSAT/Hepcidin Ratios

TSAT levels were not significantly different between the total IRIDA group and the IDA group (Figure 2a; *p* = 0.097), although the variability in the IRIDA group was higher. Hepcidin levels were significantly higher (Table 3; Figure 2b; *p* < 0.001) and TSAT/hepcidin ratios were significantly lower (*p* < 0.001; Table 3; Figure 2c) in IRIDA patients compared to IDA controls. In addition, TSAT/hepcidin ratios were significantly lower in biallelic versus monoallelic affected IRIDA patients (*p* = 0.021; Table 3; Figure 2c), and in male and premenopausal female IRIDA patients compared to their IDA counterparts (*p* = 0.002 and *p* < 0.001, respectively; Table 3).

### 2.3. Diagnostic Properties of the TSAT/Hepcidin Ratio

Receiver operating characteristic (ROC) curve analysis was performed to assess the ability of the TSAT/hepcidin ratio to differentiate between IRIDA and IDA subjects (Figure 2d). The area under the curve (AUC) for the TSAT/hepcidin ratio was 1.000 (*p* < 0.001). Using the Youden index, an optimal cut-off value of 5.6%/nM was established (Appendix A). At this cut-off point, a TSAT/hepcidin ratio of 5.6%/nM or lower distinguished IRIDA patients from IDA controls with both a specificity and a sensitivity of 100% (95% confidence interval, CI, 91–100% and 84–100%, respectively). The AUC for the TSAT/hepcidin ratio in the biallelic IRIDA group versus the IDA group was 1.000 (*p* < 0.001) with both a specificity and a sensitivity of 100% (95% CI, 91–100%, and 74–100%, respectively) at a cut-off point of 4.3%/nM. The AUC in the monoallelic affected group versus the IDA group was 1.000 (*p* < 0.001) with 100% specificity and sensitivity (95% CI, 91–100%, and 70–100%, respectively) at a cut-off point of 5.6%/nM.

### 2.4. Sensitivity Analyses

The results were materially unchanged when hepcidin levels below the lower limit of detection (LLOD) of 0.5 nM were imputed with 0.49 nM (Appendix A). In addition, the exclusion of the IDA control with a monoallelic pathogenic *TMPRSS6* variant did not result in substantial changes in the cut-off value and diagnostic accuracy of the TSAT/hepcidin ratio (Appendix A).

## 3. Discussion 

In this study, we used a standardized plasma hepcidin assay to distinguish *TMPRSS6*-related IRIDA from IDA due to other causes, provided inflammation is absent or low and no recent iron therapy has been provided. In this population, the TSAT/hepcidin ratio has an excellent discriminative performance with 100% specificity and sensitivity at a cut-off value of 5.6%/nM or lower to indicate the presence of IRIDA.

To our knowledge, there is one other group of researchers that also used the ratio between hepcidin and TSAT to differentiate between IRIDA and other forms of IDA. Heeney et al. assessed the diagnostic utility of multiple derivative hepcidin and ferritin indices, including the TSAT/log_10_(hepcidin) ratio, to predict which patients were most likely to have biallelic pathogenic *TMPRSS6* variants in a group of patients who had a high pre-test probability of having IRIDA [13]. They included chronic IDA subjects with and without pathogenic *TMPRSS6* variants with TSAT levels ≤15.0% and a poor response to at least one course of oral iron supplementation.

In agreement with their observations, we found that hepcidin levels were significantly higher in IRIDA patients compared to IDA controls. However, in our population we observed a higher ability of the TSAT/hepcidin ratio to discriminate between IRIDA and *TMPRSS6*-unrelated IDA. Heeney et al. reported an AUC of 0.886 for the TSAT/log_10_(hepcidin) ratio to differentiate between biallelic affected IRIDA patients and wild-type IDA controls with a specificity of 89% (95% CI, 75–95%) at a fixed sensitivity of 80% and a cut-off value of <4.0%/ng per mL (i.e., <1.43%/nM; molecular weight of hepcidin-25: 2.789 kDa) [13].

These differences in ROC curve analysis and cut-off values can likely be explained by the choice of distinct study populations and the use of differently calibrated hepcidin assays. We assessed the performance of the TSAT/hepcidin ratio to distinguish both biallelic and monoallelic affected IRIDA patients from controls with multi-causal IDA, while Heeney et al. used the TSAT/log_10_(hepcidin) ratio to predict the presence of biallelic pathogenic *TMPRSS6* variants in patients with chronic IDA [13]. In addition, the wild-type controls in their study population had a higher pre-test probability of having IRIDA, since a poor response to at least one course of oral iron supplementation was one of their inclusion criteria, while we did not test responsiveness to oral iron before enrollment of our IDA controls. Furthermore, the provision of recent iron therapy in their study population could have differently affected hepcidin levels and TSAT over time, and thus might have influenced the TSAT to hepcidin ratio in their control group. Last, the group of Heeney et al. did not use a calibrator that is traceable to primary reference material. The most recent hepcidin round robin (or hepcidin comparison) study showed that hepcidin levels measured by the hepcidin ELISA of Heeney et al. were approximately a factor of three higher than the levels from our assay [21]. When for comparison of results we multiply our hepcidin values by three and assess the performance of the TSAT/log_10_(hepcidin) ratio on our data, this results in an AUC of 0.991, which is close to the AUC of 1.000 as obtained in our untransformed data. We conclude that despite the use of distinct study populations and differently calibrated hepcidin assays, the ratio between TSAT and hepcidin, either as TSAT/hepcidin or as TSAT/log_10_(hepcidin), is a useful diagnostic tool to discriminate between IRIDA patients and IDA controls.

Our study has several limitations. First, in the IDA group one control had a heterozygous *TMPRSS6* variant that has been described in functional studies as being unable to suppress hepcidin expression. Taking into account all available clinical, diagnostic, and treatment characteristics of this patient (see Section 2.1.2), we considered IRIDA unlikely. In addition, exclusion of this subject in our sensitivity analysis did not result in substantial changes in the cut-off value and diagnostic performance of the TSAT/hepcidin ratio (Appendix A).

Second, comorbid conditions could have affected the performance of the TSAT/hepcidin ratio to distinguish IRIDA from IDA. However, hepcidin levels were very low in the IDA control group (i.e., <0.5 nM) despite the presence of conditions that could stimulate hepcidin production relative to TSAT (e.g., obesity and mild to moderate chronic kidney disease) [8,24,31]. These observations suggest that inhibition of hepcidin production by severe IDA trumps the stimulation of its synthesis by certain comorbid conditions, corroborating previous findings [8,18,32]. Since we excluded subjects with inflammatory conditions and subjects recently treated with iron, future studies are needed to assess the TSAT/hepcidin ratio in these patient groups. In patients with moderate-to-severe inflammation, we expect that the TSAT/hepcidin ratio has a lower specificity to distinguish IRIDA from *TMPRSS6*-unrelated IDA, as inflammation has been demonstrated to increase hepcidin synthesis and lower TSAT levels [8], which might result in TSAT/hepcidin ratios more resembling those observed in IRIDA patients. In patients recently treated with iron supplements, we hypothesize, based on previous research by Stoffel et al. [20], that >48 h after iron supplementation relative increases of TSAT and hepcidin will be similar and the TSAT/hepcidin ratio might still be able to distinguish IRIDA from *TMPRSS6*-unrelated IDA. However, it has yet to be proven whether relative increases in TSAT and hepcidin are identical in time after iron supplementation in mildly and severely iron-deficient patients, anemic patients, or IRIDA patients.

Third, the sample size was restricted and there was a significant age difference between the IRIDA group and IDA control group, in which no children were included. In healthy non-iron deficient and non-anemic adults and children, the TSAT/hepcidin ratio seems to be comparable between adults and children <12 years, with a temporary increase in the ratio after the age of 12 years [29,30]. We hypothesized that this is due to a decrease in hepcidin levels in response to the increased production of gonadal hormones during adolescence, which might reflect a regulatory mechanism to adapt to the increased iron demands needed for rapid growth during puberty and to compensate for iron losses during menstrual bleeding in girls [30]. Whether the TSAT/hepcidin ratio would be higher in adolescents with IDA compared to their non-iron deficient and non-anemic peers remains to be determined, since IDA influences both TSAT and hepcidin levels [1,2]. Interestingly, according to the literature and our data, the phenotype of many IRIDA patients tends to become milder after the onset of adolescence, which seems to persist into adulthood [10,11]. Moreover, according to our data, monoallelic IRIDA patients have a milder phenotype with an overall relatively higher TSAT/hepcidin ratio compared to biallelic IRIDA patients and tend to be diagnosed at an older age. This might lead to a higher percentage of monoallelic IRIDA patients among groups of older IRIDA patients. Therefore, including IRIDA patients of higher age might result in smaller differences in the TSAT/hepcidin ratio compared to IDA controls than observed in our study population, which should be investigated in future studies.

The strengths of our study are: (i) we confirmed the ability of the TSAT/hepcidin ratio to distinguish both monoallelic and biallelic affected IRIDA patients from controls with multi-causal IDA who had absent or low signs of inflammation and who did not receive recent iron therapy; (ii) we proposed a cut-off value for the use of the TSAT/hepcidin ratio as a diagnostic test in the workup of iron-deficient microcytic anemic patients suspected for the presence of IRIDA; (iii) we used a standardized assay for hepcidin measurements, which enables comparison with hepcidin assays used elsewhere, provided these are standardized by using the same second reference material for calibration [21,22]. This will ultimately allow global uniform decision-making based on TSAT/hepcidin ratios; (iv) we described in detail the baseline characteristics and comorbid conditions that could have influenced hepcidin production in IDA controls, which indicate that severe IDA is a stronger suppressor of hepcidin than hepcidin induction by obesity or mild to moderate chronic kidney disease.

Taken together, our findings suggest that the TSAT/hepcidin ratio is a useful tool to distinguish IRIDA early in the diagnostic workup of IDA from other causes of microcytic anemia. Our observations show that when using a standardized hepcidin assay, a TSAT/hepcidin ratio ≤5.6%/nM strongly indicates the presence of IRIDA, provided moderate-to-severe inflammation is absent and patients did not receive recent iron therapy. This could assist clinicians with referring patients suspected of having IRIDA for *TMPRSS6* analysis in a timely manner so that they can benefit from early initiation of effective treatment (usually IV iron therapy) after IRIDA has been confirmed by the presence of biallelic or monoallelic pathogenic *TMPRSS6* variants. Moreover, our results indicate that a low ratio is consistent with the diagnosis of IRIDA in non-inflamed iron-deficient patients where the *TMPRSS6* genotype is not conclusive, as is the case in iron-deficient patients with a monoallelic pathogenic *TMPRSS6* variant or with biallelic or monoallelic variants of unknown significance. Furthermore, in patients with unexplained IDA unresponsive to oral iron treatment who have a very low malignancy risk (e.g., children and young adults) and no signs of inflammatory or ulcerative disease, application of the TSAT/hepcidin ratio may prevent unnecessary invasive gastrointestinal endoscopy if patients with a ratio suspect for having IRIDA are referred for *TMPRRS6* sequencing before performing an endoscopic evaluation. This could improve effective and patient-friendly care for IRIDA patients.

Before considering the use of the TSAT/hepcidin ratio in clinical practice, it needs to be validated in IDA patients representative of those visiting secondary and tertiary healthcare centers, including children, patients with inflammatory conditions, and patients that have recently been treated with oral iron therapy.

## 4. Materials and Methods

### 4.1. IRIDA Patients

We screened the Radboud university medical center Iron Biobank, Nijmegen, Netherlands for patients who were diagnosed with IRIDA between 2010 and May 2019. This registry is a facility to optimize the use and distribution of biomaterial and/or demographic and clinical patient data for scientific research on rare inherited iron disorders [33]. We defined IRIDA probands as patients having both an ‘IRIDA phenotype’, i.e., microcytic anemia, TSAT < 10.0% (before start of parenteral iron), Hb and MCV not or partially responsive to oral iron (<2.0 g/dL Hb increment after 3 weeks of iron therapy) [25], and an ‘IRIDA genotype’, i.e., (a) monoallelic (heterozygous) or biallelic (homozygous or compound heterozygous) class 4 or 5 pathogenic *TMPRSS6* variant(s) [34]. We only included IRIDA subjects if they were living in the Netherlands and if a TSAT/hepcidin ratio was available. Patients with signs of moderate-to-severe inflammation (CRP ≥ 10.0 mg/L) or missing CRP levels were excluded to eliminate substantial hepcidin induction by inflammatory conditions.

Genotyping of IRIDA patients was performed by PCR, DNA Sanger sequencing (until March 2014), and ion torrent sequencing (after March 2014) of the exons of *TMPRSS6*, thus analyzing the coding part and the exon–intron boundaries of the gene. MLPA was conducted in patients with an IRIDA phenotype in whom only a monoallelic pathogenic *TMPRSS6* variant was found to exclude large deletions and/or duplications in the ‘healthy’ allele [11,35]. The pathogenicity of genetic variants was assessed by review of the literature on previously reported cases and functional studies, an association of the variant with the phenotype within a family, and by bioinformatic tools (SIFT, Align GVGD, Polyphen, SpliceSiteFinder-like, MaxEntScan, NNSplice, GeneSplicer, and Human Splicing Find, all as part of the Alamut^®^ Visual software) [11,36,37]. *TMPRSS6* variants were classified according to the joint English and Dutch practice guideline as either clearly pathogenic (class 5), likely to be pathogenic (class 4), uncertain pathogenicity (class 3), unlikely to be pathogenic (class 2), or clearly not pathogenic (class 1) [34].

### 4.2. IDA Controls

Subjects in the control group were prospectively enrolled in consecutive series between February 2018 and May 2019 while attending different departments of the Máxima Medical Center (MMC) Veldhoven, a large secondary healthcare center in the Netherlands, for workup of IDA. After informed consent, we evaluated their laboratory results, medical history, and medication use for study eligibility.

We sought to create a control group that resembled IRIDA patients regarding their biochemical characteristics at the time of presentation with anemia. Therefore, we included IDA controls if they had microcytic anemia (i.e., MCV ≤ 80.0 fL, Hb < 13.0 g/dL for men; <12.0 g/dL for women) with TSAT levels ≤15.0% and ferritin levels ≤40.0 µg/L. These IDA criteria were created using recommendations from the World Health Organization (WHO) [38], except with a higher cut-off value for ferritin levels than is usually seen in ID (i.e., ferritin levels <15.0 µg/L) [39]. The rationale for this is that in IRIDA patients ferritin levels are generally within the low–normal range before the start of IV iron treatment due to iron maldistribution, in which inappropriately high hepcidin levels lead to iron sequestration in macrophages [7,11,13,26]. For ethical reasons, we only included IDA controls aged ≥18 years.

Exclusion criteria were the use of either oral or parenteral iron supplementation in the past 3 months, inflammatory conditions (e.g., active inflammatory bowel disease, rheumatic disease, malignancy), CRP levels ≥10.0 mg/L, bariatric surgery, and patients suspected of having severe renal impairment (defined as an eGFR <30 mL/min/1.73 m^2^) [40], or chronic liver disease (defined as alanine aminotransferase, ALT, >40.0 IU/L), since these factors may potentially influence hepcidin production [8,17,18,20,41,42,43].

All coding exons, including the intron–exon boundary sequences of the *TMPRSS6* gene, were analyzed using an Illumina NextSeq 500 sequencer (Illumina, Inc., San Diego, CA, USA; with minimal coverage of 40 reads) after enrichment with single-molecule molecular inversion probes (smMIPs) [44], followed by Sanger sequencing for the exons and intron–exon boundaries that could not be analyzed by smMIPs. We assessed the pathogenicity of *TMPRSS6* variants using the same approach as we used to evaluate the pathogenicity of genetic variants in the IRIDA group [11], as explained above in more detail.

The study was conducted according to the principles of the Declaration of Helsinki and reported according to the Standards for Reporting of Diagnostic Accuracy (STARD) guidelines [45]. The study was approved by the accredited Medical Research Ethics Committee of the MMC and registered in the Netherlands Trial Register (NTR7023) as ‘SATURNUS Study’, an acronym for ‘Transferrin Saturation/Hepcidin ratio: a study on the diagnostic utility in the differentiation of Iron Refractory Iron Deficiency Anemia from Iron Deficiency Anemia’ [46].

### 4.3. Laboratory Measurements

For IRIDA and IDA patients, Hb, red blood cell indices, serum iron parameters, CRP, and (if applicable) serum creatinine and ALT were measured in accredited Dutch hospital laboratories. The CKD-EPI formula was used to estimate the glomerular filtration rate in patients in whom serum creatinine was measured [40].

In both IRIDA patients and IDA controls, hepcidin-25 was measured by weak cation exchange time-of-flight mass spectrometry (WCX-TOF MS) in freshly thawed heparin plasma samples that had been kept in aliquots at −80 °C for a maximum of 14 months [47]. Since February 2019, this assay has been standardized, using secondary reference material that was value assigned by provisional primary reference material [21]. Reference ranges for this standardized assay are available on our website [29]. Measurements of plasma hepcidin in the IRIDA group were performed as part of patient care and before the standardization of the assay, while those in the IDA control group were performed as a batch in May 2019 with the same assay but after standardization. Standardization slightly altered plasma hepcidin values, since standardized results were a factor 1.054 higher compared to historic results obtained without standardization [29]. The stability of the assay in time was ensured by using plasma matrix-based quality controls and Westgard rules to evaluate the results [48].

### 4.4. Statistical Analysis

Descriptive statistics were reported as medians and 25th and 75th percentiles (IQR) or ranges for continuous variables, and as absolute numbers and percentages for categorical variables, using original untransformed data. Differences in baseline characteristics between the IRIDA group and the IDA group were evaluated by the Chi-squared test for categorical variables and by the Mann–Whitney U test for continuous variables because of non-parametric data. In both IRIDA patients and IDA controls, analyses of plasma hepcidin levels and TSAT/hepcidin ratios were stratified by sex and female subjects were also stratified by menopausal status (premenopausal < 55 years and postmenopausal ≥55 years) due to sex- and age-specific differences in hepcidin levels that have been described earlier [29,49]. In addition, the IRIDA group was stratified by age in children (<18 years) and adults (≥18 years), as hepcidin reference ranges differ between both groups [29,30]. Since we considered the difference factor of 1.054 between non-standardized and standardized hepcidin levels negligible, we used uncorrected plasma hepcidin levels in our analyses. In addition, hepcidin levels below LLOD (i.e., <0.5 nM) were imputed with an average value of 0.25 nM.

Our previous work indicated that the TSAT/hepcidin ratio is a promising tool to discriminate between biallelic and monoallelic IRIDA patients and between monoallelic IRIDA patients and their phenotypically unaffected relatives without the same heterozygous pathogenic *TMPRSS6* variant [11]. Therefore, we assessed the performance of the TSAT/hepcidin ratio to distinguish IRIDA from multi-causal IDA by ROC curve analysis. We calculated the AUC and used the Youden index (sensitivity + specificity −1) to select the optimal cut-off value for the TSAT/hepcidin ratio [50]. *p*-values less than 0.05 were considered statistically significant. Since imputation of hepcidin levels with 0.25 nM may lead to an overestimation of the difference between the TSAT/hepcidin ratio of the IRIDA group and IDA control group, we performed a sensitivity analysis by imputing hepcidin levels of 0.49 nM for levels below the LLOD (Appendix A). SPSS version 22 and GraphPad Prism version 8.4.2 were used for data analysis.

## 5. Conclusions

In short, the TSAT/hepcidin ratio is a promising tool to distinguish IRIDA from other causes of microcytic anemia at an earlier stage in the diagnostic workup of IDA. This could help clinicians to prevent misdiagnosis and unnecessary invasive diagnostic tests in IRIDA patients. Therefore, future studies are warranted to explore the ratio in a broader iron-deficient population.

## Figures and Tables

**Figure 1 ijms-23-01917-f001:**
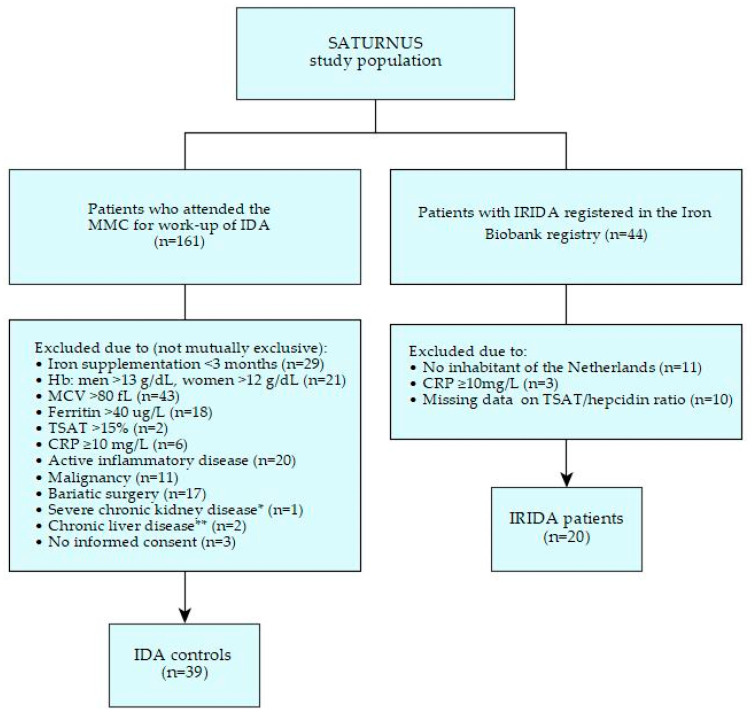
Flow chart selection process. SATURNUS, an acronym for the ‘Transferrin Saturation/Hepcidin ratio: a study on the diagnostic utility in the differentiation of Iron Refractory Iron Deficiency Anemia (IRIDA) from Iron Deficiency Anemia (IDA)’; MMC, Máxima Medical Center; Hb, hemoglobin; MCV, mean corpuscular volume; TSAT, transferrin saturation; CRP, C-reactive protein. * Severe chronic kidney disease, estimated Glomerular Filtration Rate (eGFR) < 30 mL/min/1.73 m^2^, ** Chronic liver disease, cirrhosis, or alanine aminotransferase (ALT) > 40 U/L.

**Figure 2 ijms-23-01917-f002:**
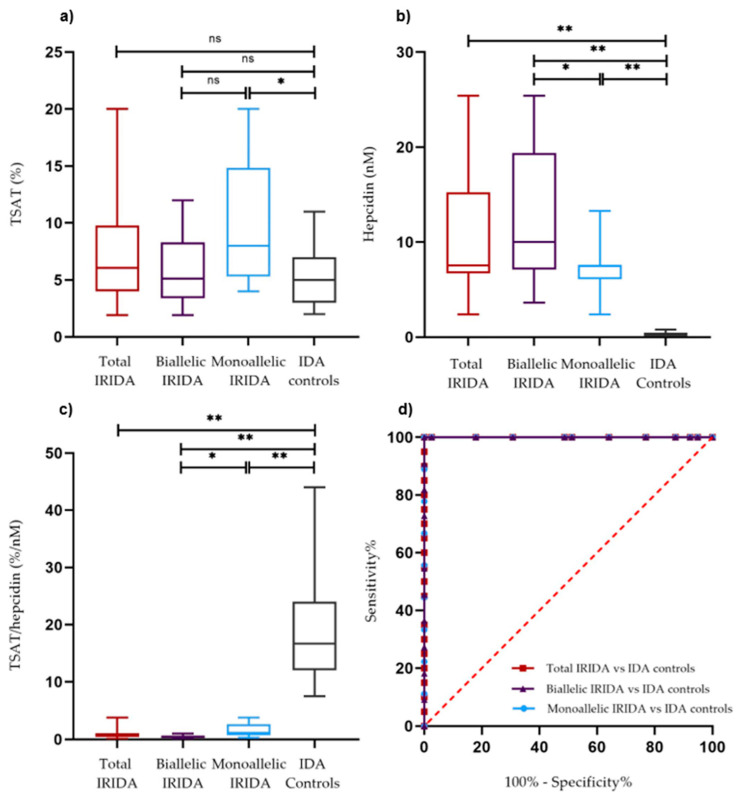
(**a**–**c**) Transferrin saturation (TSAT), plasma hepcidin levels, and TSAT/hepcidin ratio in the total IRIDA group (*n* = 20), biallelic IRIDA patients (*n* = 11), monoallelic IRIDA patients (*n* = 9) and IDA controls (*n* = 39). Box and whisker plots present the quartiles (box), the medians (bold line), and the minimum and maximum (whiskers). (**d**) Receiver Operating Characteristic (ROC) curve analysis (see also Appendix A) comparing the TSAT/hepcidin ratio (%/nM) in IDA controls versus the total IRIDA group (red), versus the biallelic IRIDA group (purple), and versus the monoallelic IRIDA group (blue). Ns, not significant; * = *p* < 0.05; *** = p <* 0.001, by non-parametric Mann–Whitney U test. AUC, area under the curve.

**Table 1 ijms-23-01917-t001:** Clinical, biochemical, genetic, and treatment characteristics of patients with IRIDA and controls with iron deficiency anemia due to other reasons.

	IRIDA Patients (*n* = 20)			IDA Controls (*n* = 39)		
Characteristics	N	Median or Count (%)	IQR	N	Median or Count (%)	IQR
Age at presentation with anemia, years	20			N/A	N/A	N/A
Total group		10	2–31			
Biallelic IRIDA		4	2–10			
Monoallelic IRIDA		31	10–43			
Age at TSAT/hepcidin assessment, years	20			39		
Total group		25	7–40		62	52–71
Biallelic IRIDA		9	6–21		N/A	N/A
Monoallelic IRIDA		40	31–48		N/A	N/A
Women	20	15 (75)	N/A	39	29 (74)	N/A
Systemic disease	20		N/A	39		N/A
Diabetes mellitus		1 (5)			12 (31)	
Hypothyroidism		NP			4 (10)	
IBD (complete remission)		NP			1 (3)	
Rheumatic disease (complete remission)		NP			1 (3)	
No systemic disease		19 (95)			21 (54)	
Hb, g/dL ^a^	19	11.4	9.8–12.0	39	9.3	4.7–6.5
MCV, fL ^a^	19	70.0	63.0–80.0	39	73.0	66.0–77.0
Ferritin, µg/L ^a^	19	130.0	33.0–293.0	39	9.0	6.0–14.0
TSAT, % ^a^	20	6.0	4.0–9.8	39	5.0	3.0–7.0
CRP, mg/L ^a^	20	<5.0	<5.0–<5.0	39	1.8	0.7–2.5
Pathogenic *TMPRSS6* variant (s)	20		N/A	36		N/A
Biallelic		11 (55)			NP	
Monoallelic		9 (45)			1 (3)	
No pathogenic variants		NP			35 (97)	

^a^ At time of TSAT/hepcidin assessment, after oral or oral and parenteral iron therapy in IRIDA patients and before initiation of iron supplementation in IDA controls. IRIDA, iron refractory iron deficiency anemia; IDA, iron deficiency anemia; IQR, interquartile range; N/A, not applicable; NP, not present; TSAT, transferrin saturation; IBD, inflammatory bowel disease; Hb, hemoglobin; MCV, mean corpuscular volume; CRP, C-reactive protein.

**Table 2 ijms-23-01917-t002:** Additional characteristics of IDA controls.

	IDA Controls (*n* = 39)		
Characteristics	N	Median or Count (%)	IQR
**ALT, U/L**	21	18.0	16.0–25.0
**eGFR (CKD-EPI), mL/min/1.73 m^2 a^**	33	89	65–≥90
BMI, kg/m^2^	39	28	24–34
Normal weight (18.5–24.9)		11 (28)	
Overweight (25–29.9)		9 (23)	
Obese (≥30)		17 (44)	
Unknown		2 (5)	
Assessment of underlying disorder of IDA	39		N/A
Gastrointestinal bleeding		12 (31)	
Gynecological bleeding		5 (13)	
Malabsorption		4 (10)	
Unexplained IDA		18 (46)	
Assessment of medication use	39		N/A
Anticoagulants		6 (15)	
Antithrombotic agents		12 (31)	
NSAIDs		1 (3)	
Proton pump inhibitors		14 (36)	
Corticosteroids		1 (3)	
Assessment of response to iron therapy in unexplained IDA ^b^	18		N/A
Hb increase < 2.0 g/dL after oral iron		1 (6)	
Hb increase < 2.0 g/dL after IV iron		1 (6)	
Hb increase ≥ 2.0 g/dL after oral iron		4 (22)	
Hb increase ≥ 2.0 g/dL after IV iron		9 (50)	
No iron supplementation		3 (17)	

^a^ eGFR was ≥90 mL/min/1.73 m^2^ in 49% (*n* = 16), 60–89 mL/min/1.73 m^2^ in 30% (*n* = 10), 45–59 mL/min/1.73 m^2^ in 18% (*n* = 6) and 30–44 mL/min/1.73 m^2^ in 3% (*n* = 1). ^b^ Response to iron therapy 3 weeks after initiation of iron supplementation. IDA, iron deficiency anemia; IQR, interquartile range; N/A, not applicable; ALT, alanine aminotransferase; eGFR, estimated Glomerular Filtration Rate; BMI, body mass index; Hb, hemoglobin, NSAIDs, non-steroidal anti-inflammatory drugs; IV, intravenous.

**Table 3 ijms-23-01917-t003:** Hepcidin levels and TSAT/hepcidin ratios in IRIDA patients and IDA controls.

	IRIDA Patients			IDA Controls				Reference Range (General Population) ^a^	
	N	Median	IQR	N	Median	IQR	Mann–Whitney U Test *p*-Value	Median	P2.5–P97.5
Hepcidin level (nM) ^b^									
Total group	20	7.6	6.7–15.3	39	0.3	0.3–0.3	<0.001	N/A	N/A
Men	5	7.6	7.1–19.2	10	0.3	0.3–0.3	<0.001	4.7	<0.5–15.5
Premenopausal women (age < 55 years)	14	7.6	6.9–14.2	10	0.3	0.3–0.3	<0.001	2.1	<0.5–13.0
Postmenopausal women (age ≥ 55 years)	1	2.4	N/A	19	0.3	0.3–0.3	N/A	5.2	<0.5–16.5
Adults	12	7.5	6.0–12.5	39	0.3	0.3–0.3	<0.001	N/A	N/A
Children (<18 years)	8	7.7	7.2–20.9	N/A	N/A	N/A	N/A	1.5	0.1–9.8
Biallelic pathogenic *TMPRSS6* variants ^c^	11	10.0	7.1–19.4	N/A	N/A	N/A	N/A	N/A	N/A
Monoallelic pathogenic *TMPRSS6* variants ^c^	9	7.5	6.1–7.6	N/A	N/A	N/A	N/A	N/A	N/A
TSAT/hepcidin ratio (%/nM)									
Total group	20	0.6	0.4–1.1	39	16.7	12.0–24.0	<0.001	N/A	N/A
Men	5	0.5	0.2–1.1	10	20.0	15.0–34.0	0.002	6.9	1.6–243.0
Premenopausal women (age < 55 years)	14	0.7	0.5–1.1	10	16.0	8.0–24.0	<0.001	13.2	1.9–312.9
Postmenopausal women (age ≥ 55 years)	1	3.8	N/A	19	20.0	12.0–24.0	N/A	5.4	1.4–69.6
Adults (≥18 years)	12	0.9	0.5–2.2	39	16.7	12.0–24.0	<0.001	N/A	N/A
Children (<18 years)	8	0.5	0.3–0.7	N/A	N/A	N/A	N/A	15.4	1.4–665.5
Biallelic pathogenic *TMPRSS6* variants ^d^	11	0.5	0.3–0.6	N/A	N/A	N/A	N/A	N/A	N/A
Monoallelic pathogenic *TMPRSS6* variants ^d^	9	1.1	0.7–2.6	N/A	N/A	N/A	N/A	N/A	N/A

^a^ Reference values for serum hepcidin-25 and TSAT/hepcidin ratio are available [29,30]. ^b^ Hepcidin levels below the lower limit of detection (<0.5 nM) were imputed with an average value of 0.25 nM. ^c^ Hepcidin levels in biallelic versus monoallelic IRIDA patients: *p* = 0.044, Mann–Whitney U test. ^d^ TSAT/hepcidin ratio in biallelic versus monoallelic IRIDA patients: *p* = 0.021, Mann–Whitney U test. IQR, interquartile range; N/A, not applicable.

## Data Availability

The data presented in this study are available upon request from the corresponding author (D.W.S.). The data are not publicly available due to ethical/privacy restrictions.

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
