# Peer review of "Transferrin Saturation/Hepcidin Ratio Discriminates TMPRSS6-Related Iron Refractory Iron Deficiency Anemia from Patients with Multi-Causal Iron Deficiency Anemia"

_ijms, 2022, doi:10.3390/ijms23031917_

Round 1

Reviewer 1 Report

Thanks for your submission. I have several comments that I would like you to consider

-On page 2, line 60, I assume that the patients that present at an older age have microcytic anemia as well? The text says 'regarding hemoglobin and MCV'.

-Why were subjects only recruited from the Netherlands?

-Did you consider that your subjects may have mutations in other iron homeostasis genes? Could these mutations impact your study?

-On page 6, in the first full paragraph, you describe an IDA control patient that carried a TMPRSS6 mutation. Please indicate at this point that the patient was retained in the study group (you did indicate this later on in the discussion).

Reviewer 2 Report

I have read an interesting paper entitled “Transferrin Saturation/Hepcidin ratio discriminates TMPRSS6-2 related Iron Refractory Iron Deficiency Anemia from Patients 3 with multi-causal Iron Deficiency Anemia”. This is an extremely well-written paper, with high scientific value, from my point of view.

Here are comments:

- if you have various ideas from the same reference (lines 62-68) you can put it at the end of the exposed ideas

- please explain the age difference between the two groups – why you decided not to include patients under 18 years in IDA group?

- maybe it will be useful to present the limitations of the study in a more compact way, so as not to diminish its value. Now, explanations about the limitations of the study are much more elaborate compared to the discussion section itself.

- in the discussion section it would be expected that there would be some elaborate discussion of the demographic characteristics of the two groups, given that in the results section much emphasis was placed on them

- section 4 should be named “Discussion and conclusion”

Reviewer 3 Report

The publication submitted in IJMS by van der Staaij and colleagues entitled “Transferrin Saturation/Hepcidin ratio discriminates TMPRSS6-related Iron Refractory Iron Deficiency Anemia from Patients with multi-causal Iron Deficiency Anemia” sent to me for review tries to use the perfectly developed by Prof Swinkels team method of measuring hepcidin in order to diagnose IRIDA and distinguish this anemia from IDA (multi casual forms). This work is a kind of continuation of research aimed at accurate diagnosis of various types of anemia (Donker et al., 2015, 2016).

It is true that the use of an indicator/factor based on TSAT/hepc is not a new idea, as it is already proposed by the group from Boston (Heeyes et al., 2018), however, the use of WCX-TOF MS is, according to my knowledge, much more reliable than ELISA. For this reason, in the introduction, it is worth mentioning the publications where the authors used the above-mentioned method to measure hepcidin in various species (data from the website hepcidin.com).

In contrast to the precursors of this type of measurement, the results of discrimination between IRIDA and IDA are much better. As stated in the authors' discussion, the use of a different hepcidin calibration may contribute to 100% detection of IRIDA, among other anemias.

Of course, the differences may also result from the low variability of the population taken for the study in van der Staaij and Heeney, 20 versus 44 respectively. Additionally, the lack of children in the IDA group may affect the flattening of the variability of the results.

Fortunately, the authors take into account all the shortcomings of their research, which are reflected in quite a long discussion.

The authors are also confident in their strengths, which certainly is to avoid unnecessary invasive examination in children or the elderly.

Therefore, I share the authors' enthusiasm as well as their objection related to the use of TSAT / hepc in clinics.  

As a reviewer, I also point out some editorial shortcomings:

Table 1a and b. Incomprehensible, unreadable. Maybe it is worth introducing line shading in the tables?

Line 297-300. A bold assumption. In my country, young people are increasingly obese compared to 50 years old. But in general, young people should be slimmer and more fit and athletic.

Line 146, space

In general, I believe that the work meets the publication requirements in IJMS and should be published after minor changes.

Regards.

Reviewer

The publication submitted in IJMS by van der Staaij and colleagues entitled “Transferrin Saturation/Hepcidin ratio discriminates TMPRSS6-related Iron Refractory Iron Deficiency Anemia from Patients with multi-causal Iron Deficiency Anemia” sent to me for review tries to use the perfectly developed by Prof Swinkels team method of measuring hepcidin in order to diagnose IRIDA and distinguish this anemia from IDA (multi casual forms). This work is a kind of continuation of research aimed at accurate diagnosis of various types of anemia (Donker et al., 2015, 2016).

It is true that the use of an indicator/factor based on TSAT/hepc is not a new idea, as it is already proposed by the group from Boston (Heeyes et al., 2018), however, the use of WCX-TOF MS is, according to my knowledge, much more reliable than ELISA. For this reason, in the introduction, it is worth mentioning the publications where the authors used the above-mentioned method to measure hepcidin in various species (data from the website hepcidin.com).

In contrast to the precursors of this type of measurement, the results of discrimination between IRIDA and IDA are much better. As stated in the authors' discussion, the use of a different hepcidin calibration may contribute to 100% detection of IRIDA, among other anemias.

Of course, the differences may also result from the low variability of the population taken for the study in van der Staaij and Heeney, 20 versus 44 respectively. Additionally, the lack of children in the IDA group may affect the flattening of the variability of the results.

Fortunately, the authors take into account all the shortcomings of their research, which are reflected in quite a long discussion.

The authors are also confident in their strengths, which certainly is to avoid unnecessary invasive examination in children or the elderly.

Therefore, I share the authors' enthusiasm as well as their objection related to the use of TSAT / hepc in clinics.  

As a reviewer, I also point out some editorial shortcomings:

Table 1a and b. Incomprehensible, unreadable. Maybe it is worth introducing line shading in the tables?

Line 297-300. A bold assumption. In my country, young people are increasingly obese compared to 50 years old. But in general, young people should be slimmer and more fit and athletic.

Line 146, space

In general, I believe that the work meets the publication requirements in IJMS and should be published after minor changes.

Regards.

Reviewer

Round 2

Reviewer 1 Report

All issues have been addressed!